## Replications

 

**Cite this article:** van den Broeke EN, Vanmaele T, Mouraux A, Stouffs A, Biurrun-Manresa J, Torta DM. 2021 Perceptual correlates of homosynaptic long-term potentiation in human nociceptive pathways: a replication study. *R. Soc. Open Sci.* **8**: 200830.

neuroscience/physiology

long-term potentiation, high-frequency stimulation, homotopic hyperalgesia

**Author for correspondence:**
E. N. van den Broeke
e-mail: emanuel.vandenbroeke@uclouvain.be

# Perceptual correlates of homosynaptic long-term potentiation in human nociceptive pathways: a replication study

E. N. van den Broeke[1], T. Vanmaele[2], A. Mouraux[1], A. Stouffs[1], J. Biurrun-Manresa[3] and D. M. Torta[2]

[1]Institute of Neuroscience, IoNS, Faculty of Medicine, UC Louvain, Avenue Mounier 53, B-1200, Brussels, Belgium
[2]Health Psychology Research Group, Faculty of Psychology and Educational Sciences, KU Leuven, Tiensestraat 102, B-3000 Leuven, Belgium
[3]Institute for Research and Development in Bioengineering and Bioinformatics (IBB-CONICET-UNER), National University of Entre Rios, Oro Verde, Argentina

DMT, 0000-0002-3499-3982

Animal studies have shown that high-frequency stimulation (HFS) of peripheral C-fibres induces long-term potentiation (LTP) within spinal nociceptive pathways. The aim of this replication study was to assess if a perceptual correlate of LTP can be observed in humans. In 20 healthy volunteers, we applied HFS to the left or right volar forearm. Before and after applying HFS, we delivered single electrical test stimuli through the HFS electrode while a second electrode at the contra-lateral arm served as a control condition. Moreover, to test the efficacy of the HFS protocol, we quantified changes in mechanical pinprick sensitivity before and after HFS of the skin surrounding both electrodes. The perceived intensity was collected for both electrical and mechanical stimuli. After HFS, the perceived pain intensity elicited by the mechanical pinprick stimuli applied on the skin surrounding the HFS-treated site was significantly higher compared to control site (heterotopic effect). Furthermore, we found a higher perceived pain intensity for single electrical stimuli delivered to the HFS-treated site compared to the control site (homotopic effect). Whether the homotopic effect reflects a perceptual correlate of homosynaptic LTP remains to be elucidated.

# 1. Introduction

Animal studies have shown that high-frequency electrical stimulation (HFS) of the sciatic nerve of rats induces homosynaptic long-term potentiation (LTP) between peripheral C-fibres and lamina I spinal neurons projecting to the brain [1,2]. This finding has led to the idea that LTP within spinal nociceptive pathways could be one of the processes underlying long-lasting hyperalgesia. In a first attempt to provide translational evidence for the role of LTP in hyperalgesia, Klein *et al*. [3] applied HFS to the human skin using an electrode designed to preferentially activate skin nociceptors. They found that the painful percept elicited by single electrical test stimuli delivered through the HFS electrode was enhanced for at least 60 min (further referred to as homotopic effect) after applying HFS. Moreover, they observed a long-lasting increase in mechanical pinprick sensitivity to stimuli applied on the skin surrounding the HFS-conditioned site, as well as painful sensations in response to dynamic mechanical tactile stimuli (further referred to as heterotopic effects). The authors suggested that at least part of the homotopic effect reflects a perceptual correlate of homosynaptic LTP, whereas the heterotopic effects represent a perceptual correlate of heterosynaptic LTP [3].

After this original report, the authors published a series of studies in which they consistently replicated their initial findings [4–10]. At the same time, other independent research groups also started applying HFS in humans. Contrary to Klein and colleagues, they were not able to replicate all HFS conditioning effects. Indeed, whereas the increase in mechanical pinprick sensitivity surrounding the site of HFS has been consistently observed [11–35], the increase in perceived pain elicited by single electrical stimuli (homotopic effect) has not [11,12], or has been observed to a lesser extent [35].

The aim of this study was to assess the presence of a long-lasting increase in pain perception elicited by single electrical test stimuli delivered at the conditioned site (primary outcome) as a perceptual correlate of homosynaptic LTP in human nociceptive pathways.

# 2. Materials and methods

## 2.1. Participants

Following the approval of the ethical commission (SMEC, KU Leuven: G-2020 03 1999), the experiments were performed on 20 healthy volunteers aged between 18 and 40 years old (10 women and 10 men). Participants received either course credits or monetary compensation for participating in the study. Experimental procedures were explained to each participant and written informed consent was obtained. Participants were not allowed to take part in the experiment if they reported known illnesses (e.g. including heart and vascular, respiratory and neurological diseases, presence of a pacemaker or other electronic implant, hearing and/or eye problems, psychiatric diseases), pain (e.g. acute pain at the beginning of the experiment, chronic pain of a duration of 3 months or longer), current sleep deprivation (less than 6 slept hours the night prior to the experiment), pregnancy, regularly used medication (except oral contraceptives) or drugs, use of anti-inflammatory medication and/or painkillers less than 12 h before the experiment, or if they had already participated in a study using HFS or low-frequency stimulation (LFS) [36].

## 2.2. Sample size calculation

Studies attempting to replicate a significant effect using the same sample size as the original study risk being underpowered, due to several biasing factors [37]. Thus, specific care must be taken when calculating sample size for replication studies. A novel proposal defines the concept of replication success by considering the effect and sample sizes of both original and replication studies [38]. Using the framework provided in that study, it is possible to calculate the relative sample size c (defined as the ratio between the replication sample size $n_r$ and the original sample size $n_o$) required to obtain replication success with 80% predictive power at level $\alpha = 5\%$, as a function of the two-sided $p$ value $p_o$ of the original study. In this case, $p_0 = 0.0023$ and $n_o = 7$ (for the homotopic analysis, see table 1 in [3]), which resulted in a relative sample size c = 2.5. Therefore, we recruited 20 participants.

## 2.3. Study design

The summary of the design is shown in figure 1. HFS was applied to the left or right volar forearm (5 cm from the cubital fossa) using a multi-pin electrode designed to preferentially activate nociceptors. A second

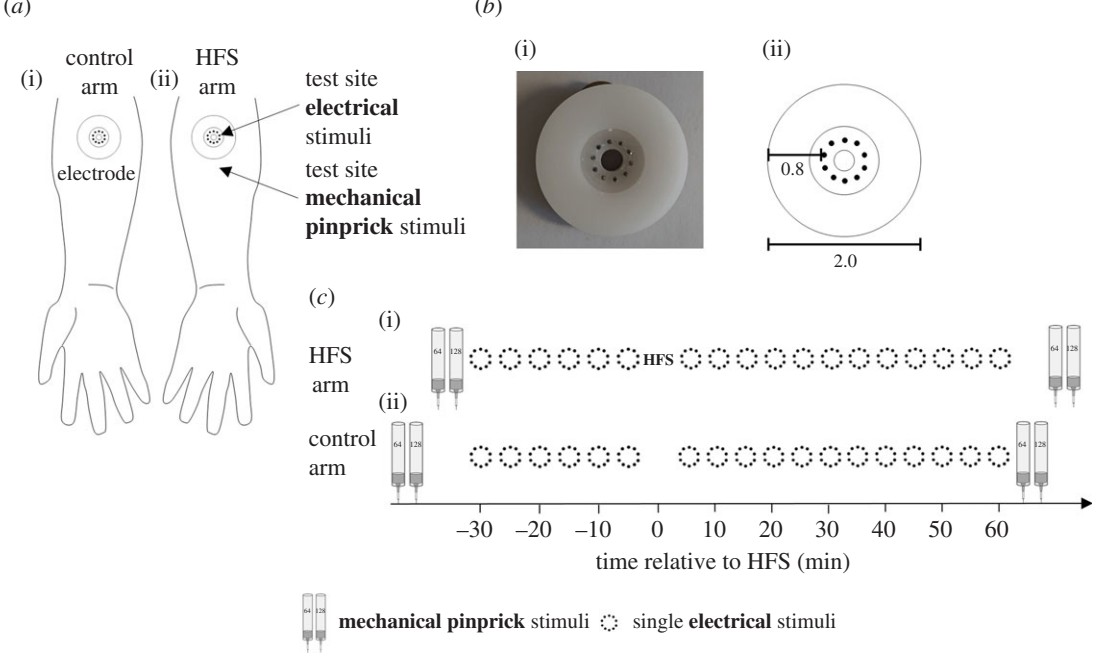

**Figure 1.** Design of the study. (*a*) HFS was applied to the left or right volar forearm skin 5 cm from the cubital fossa using a multi-pin electrode designed to preferentially activate nociceptors. An identical second electrode was attached to the skin 5 cm from the wrist. This site served as control site as no HFS was applied through this electrode. (*b*) Characteristics of the electrode. (*c*) Timeline of the experiment.

identical multi-pin electrode was placed at the same location of the contra-lateral arm but no HFS was delivered (control condition). Before and after applying HFS, single electrical test stimuli were delivered through both the HFS electrode and the control electrode in alternating order (homotopic effect). Before and 60 min after applying HFS, mechanical pinprick stimuli (64 and 128 mN) were delivered to the skin surrounding both electrodes to verify the increase in mechanical pinprick pain (heterotopic effect). At the end of the experiment, we debriefed participants and asked about their expectations.

Since our aim was to replicate the homotopic effect of HFS, we reduced the number of tests for the heterotopic effects as compared to the study of Klein *et al*. In fact, we only tested mechanical pinprick sensitivity (secondary outcome) at baseline, before applying the electrical test stimuli, and at 60 min after applying HFS, after the last electrical test stimulus. The increase in mechanical pinprick sensitivity after HFS was used to confirm the efficacy of the HFS protocol. Moreover, we aimed at preventing a potential confounding effect. Indeed, the increase in perceived intensity elicited by the mechanical stimuli is usually larger (and more pronounced) compared to the increase in perceived intensity elicited by the electrical test stimuli [35]. Alternating between mechanical stimuli and electrical ones may bias the judgement of intensity of electrical stimuli towards higher ratings, although available evidence seems to advocate against this possibility [7]. Notably, previous studies have shown that the increase in pinprick mechanical sensitivity has a half-life time of 4–5 h [9], meaning that if heterotopic effects were successfully induced they would be still observed 60 min after the HFS procedure.

## 2.4. High-frequency electrical stimulation

HFS consisted of five trains of 100 Hz electrical stimuli (square wave pulses with a pulse width of 2 ms) that lasted 1 s each and were delivered with a 10 s inter-train interval (9 s between each train). The total duration of the HFS procedure was 50 s. The trains were generated using a constant current stimulator (DS5, Digitimer Ltd, Welwyn Garden City, UK) and delivered to the forearm skin using a multi-pin electrode consisting of a ring of 10 blunt stainless steel pins (250 µm diameter each) that served as the cathode. Both the electrode through which the HFS was delivered and the electrode at the contra-lateral arm were attached to the skin using double-adhesive tape. A large surface electrode (PALS platinum 5 × 9, Axelgaard Electrical Stimulation Electrodes, Digitimer, Hertfordshire, UK), which served as the anode, was attached onto the skin of the ipsilateral upper arm (biceps). The HFS trains

were delivered at an intensity corresponding to 10 times the electrical detection threshold to a single electrical stimulus (see Experimental procedure).

## 2.5. Test stimuli

### 2.5.1. Homotopic effect: single electrical test stimuli

To assess the presence of a long-lasting homotopic increase in pain sensitivity to the electrical stimuli after HFS, single electrical test stimuli (square wave pulses with a pulse width of 2 ms) were delivered through the HFS and control electrodes before and after HFS. At each time point (every 5 min), one single electrical pulse was delivered at an intensity of 10 times the electrical detection threshold. The ratings were collected after each stimulus (see pain ratings).

### 2.5.2. Heterotopic effect: mechanical pinprick stimuli

To assess the presence of the heterotopic increase in mechanical pinprick sensitivity after HFS, static mechanical pinprick stimuli (64 and 128 mN) were applied to the skin surrounding both electrodes using calibrated pinprick stimulators (The Pin Prick, MRC Systems GmbH, Heidelberg, Germany). In the original study, seven pinprick intensities (8, 16, 32, 64, 128, 256 and 512 mN) were used. However, because we and others [11–35] were able to replicate the increased mechanical pinprick sensitivity after HFS, we decided to use only two pinprick stimulation intensities here (64 and 128 mN), instead of seven. A total of three pinprick stimulations lasting 1 s each were delivered to the skin, with the hand-held stimulator maintained approximately perpendicular to the skin. The mechanical pinprick stimuli were applied within a circular area of 15 mm from the circle of pins of the cathodal electrode and never twice at the same location to avoid sensitization of the skin due to repeated stimulation.

### 2.5.3. Pain ratings

Participants were asked to rate the magnitude of pain elicited by the electrical and mechanical test stimuli on a numerical rating scale (NRS) ranging from 0 (non-painful) to 100 (most intense pain imaginable). This rating scale was identical to the one previously used by [3–11,35]. Importantly, participants were instructed to distinguish painful from non-painful sensations by the presence of a sharp or slightly pricking or burning sensation. Participants were also instructed to pay attention to any subtle change in the sensation and were free to use integers as well as fractions.

## 2.6. Experimental procedure

Upon arrival in the laboratory, participants were informed about the procedures used in the experiment and signed the exclusion criteria list and the informed consent form. The experiment took place in a light and temperature-controlled room, where participants sat comfortably on an office chair, with their arms in a position with the palm-up on a table. Each participant was first familiarized with the experimental procedures by receiving a description of the general set-up and the stimuli that they would receive. After that, the baseline measurement for mechanical pinprick testing was performed. Then, the electrical detection threshold was determined at both electrodes separately by means of a staircase procedure with three ascending and descending staircases of single stimuli (2 ms pulse width). The first site at which the detection threshold was established (control or HFS arm) was counterbalanced across participants. The final electrical detection threshold was the geometric mean of the three series [8]. Then, the baseline measurements were performed. The electrical test stimuli were delivered to the conditioned and contra-lateral control site in alternating order within an interval of 5 min. After 30 min of baseline measurements, HFS was applied, and immediately after that, the testing continued for another 60 min in the same way as during the baseline measurements. At 60 min after applying HFS, the perceived intensity to mechanical pinprick stimuli was re-assessed.

## 2.7. Statistical analysis

The data were first checked for the assumption of normality. In the case of violation, it was log-transformed [3–11]. If there were any zeros in the ratings we added a constant value of 0.1 to all ratings in order to not

lose any values [3–11]. The ratings to electrical and pinprick stimuli were examined using a repeated measures analysis of covariance (RM ANCOVA), with Time (at 5, 10, 15, 20, 30, 35, 40, 45, 50, 55, 60 min) and Test Site (conditioned site, control site) as factors, where the covariate was the average baseline (−30–0 min) pain rating, separately for each site, for each subject. *Post hoc* tests involved all comparisons. The significance level was set at $p < 0.05$ for all statistical tests.

Since our statistical approach differed from that of the original paper, we performed a multiverse analysis. This approach involves performing different analysis across reasonable choices for data processing and displays how stable and robust the findings are [39].

# 3. Results

Twenty participants were recruited (10 females, 10 males) with a mean (±s.d., min–max) age of 22.35 (1.76, 19–27). The electrical thresholds obtained at each arm were $0.12 \pm 0.05$ (mean ± s.d.) mA for the left arm and $0.11 \pm 0.04$ mA for the right arm. We did not observe a significant difference in thresholds between the two arms (paired *t*-test: $t_{19} = 1.504$, $p = 0.149$).

## 3.1. Primary outcome: perceived pain intensity elicited by single electrical stimuli after high-frequency stimulation

Figure 2*a* shows the individual pain ratings in response to single electrical stimuli obtained before and after HFS conditioning at the HFS-treated site and control site, along with their median and interquartile ranges. One participant rated the single electrical stimuli both before and after HFS between 80 and 90 out of 100, which is much higher than the ratings of all other participants. An outlier analysis (Grubbs' test) confirmed that these ratings were statistically significant outliers compared to the ratings of the other participants. Therefore, figure 2*b* shows the median and interquartile ranges of the perceived pain intensity without the outlier.

The results of two different statistical procedures were compared regarding the primary outcome in a multiverse analysis (figure 3); the repeated measures ANCOVA (as originally planned) and another analysis similar to the one conducted in Klein *et al.* [3]. In that paper, the authors performed a $2 \times 2$ ANOVA with the factors 'site' (control versus HFS) and 'stimulation intensity' (HFS delivered at 10 versus 20 times the electrical detection threshold HFS) as independent variables. However, in the present study, we only applied one-stimulation intensity during HFS, which leaves us with the factor site only. For this reason, the $2 \times 2$ ANOVA was replaced by a paired *t*-test on the mean post-HFS pain ratings of the two sites (control versus HFS).

### 3.1.1. Repeated measures analysis of covariance

We conducted four different ANCOVAs (figure 3): with and without log-transformation of the pain ratings and with and without the outlier. Table 1 shows the main results of all four analyses. Figure 4 shows the estimated marginal means (and 95% CI) of the perceived pain intensity elicited from both sites after HFS and corrected for pre-existing baseline differences.

All analyses showed significant main effects of SITE and TIME but no significant SITE × TIME interaction. This means that, when pain ratings were averaged across time points, the perceived pain intensity elicited by the single electrical stimuli was significantly higher at the HFS-treated site compared to the control site. Moreover, when pain ratings were averaged across sites, the perceived pain intensity was different between some of the time points. *Post hoc* tests showed that the perceived pain intensity between 10 and 25 min was higher compared to 40–60 min (table 1).

### 3.1.2. Paired *t*-test

The paired *t*-test conducted on the post-HFS log-transformed pain ratings of both sites showed a statistically significant difference (table 1), indicating that the mean log-pain rating after HFS was significantly higher at the HFS site compared with the control site. *Post hoc* paired *t*-tests (uncorrected)

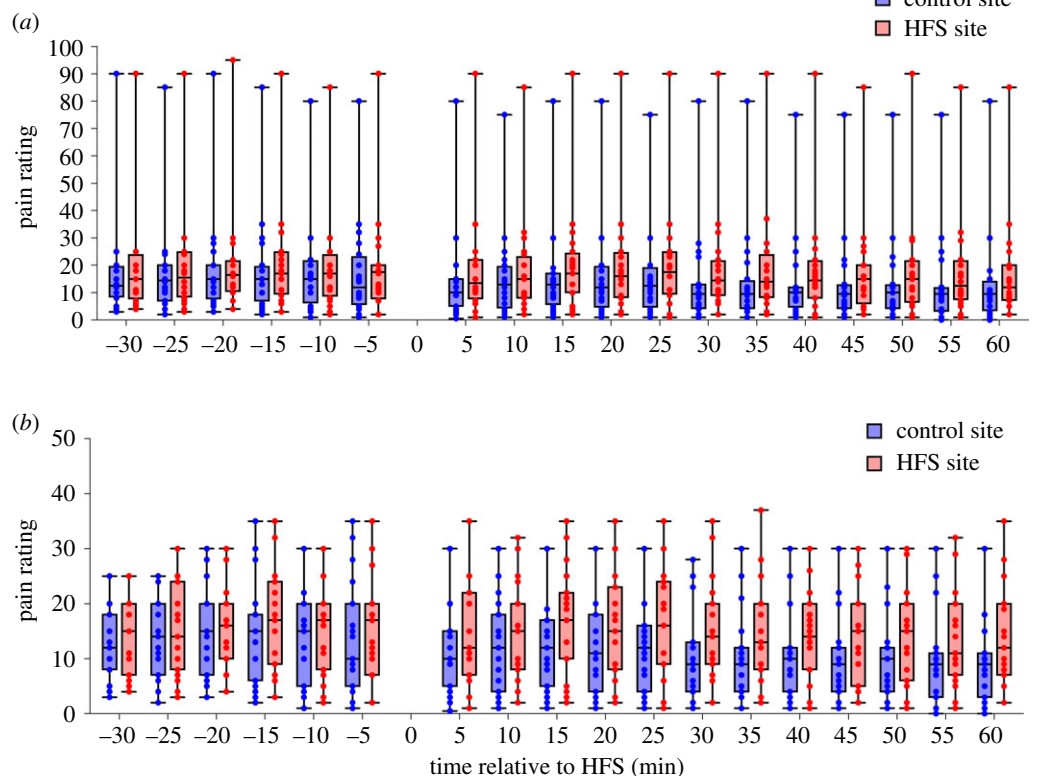

**Figure 2.** (*a*) Median and interquartile ranges (and min–max value) of the perceived pain intensity elicited by single electrical stimuli before (−30–0 min) and after HFS (5–60 min) at the site at which HFS was delivered (HFS site) and control site (control site). Dots represent individual ratings. (*b*) The same data without outlier.

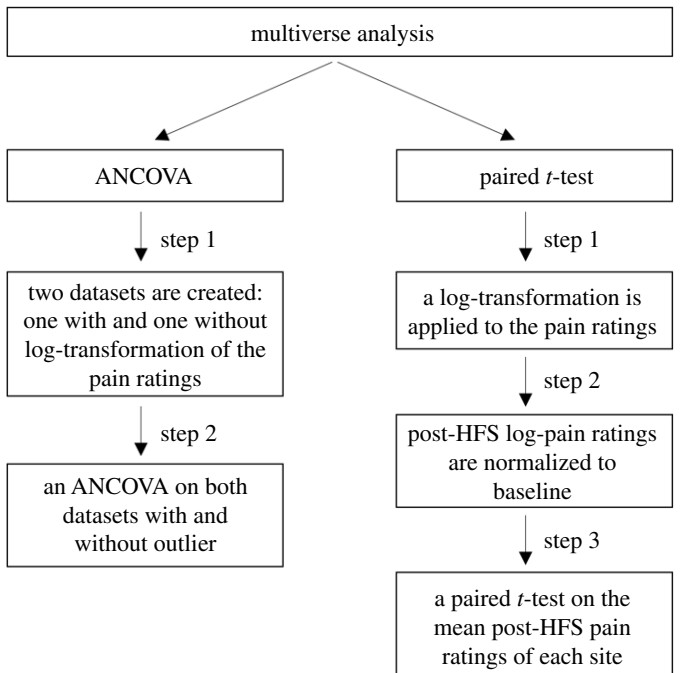

**Figure 3.** Multiverse analysis. The results of two different statistical approaches were compared: a repeated measures ANCOVA (as originally planned) and a paired *t*-test (an analysis similar to the one used in Klein *et al*. [3]). Repeated measures ANCOVA. Four different ANCOVAs were conducted: with and without log-transformation of the ratings and with and without outlier. Paired *t*-test. Before carrying out a paired *t*-test on the mean post-HFS ratings of the two sites (control versus HFS), the observed pain ratings were log-transformed and normalized by dividing the post-HFS pain ratings to the mean baseline (−30–0 min). To test at which time points the mean pain rating was significantly different between the two sites, *post hoc* paired *t*-tests (uncorrected) were conducted for each post-HFS time point.

**Table 1.** Results of the multiverse analysis for the pain ratings elicited by single electrical stimuli. * Corrected for multiple comparisons (Sidak).

| ANCOVA | | observed pain ratings with outlier | observed pain ratings without outlier | log-transformed pain ratings with outlier | log-transformed pain ratings without outlier |
|---|---|---|---|---|---|
| main effect of SITE | | $F_{1,436} = 116.123$, $p < 0.001$, partial $\eta^2 = 0.210$ | $F_{1,413} = 113.807$, $p < 0.001$, partial $\eta^2 = 0.216$ | $F_{1,436} = 83.194$, $p < 0.001$, partial $\eta^2 = 0.160$ | $F_{1,413} = 83.278$, $p < 0.001$, partial $\eta^2 = 0.168$ |
| | difference between sites (mean + 95% CI) | 3.148 (2.574–3.723) | 3.090 (2.520–3.659) | 0.121 (0.095–0.147) | 0.127 (0.100–0.154) |
| main effect of TIME | | $F_{11,436} = 5.731$, $p < 0.001$, partial $\eta^2 = 0.126$ | $F_{11,413} = 5.637$, $p < 0.001$, partial $\eta^2 = 0.131$ | $F_{11,436} = 5.978$, $p < 0.001$, partial $\eta^2 = 0.131$ | $F_{11,413} = 6.002$, $p < 0.001$, partial $\eta^2 = 0.138$ |
| | significant ($p < 0.05$)* comparisons time points (min) | 15 versus 40, 45, 50, 55 20 versus 45, 50, 55, 60 25 versus 40, 45, 50, 55, 60 | 15 versus 40, 45, 50, 55, 60 20 versus 45, 50, 55, 60 25 versus 40, 45, 50, 55, 60 | 10 versus 55, 60 15 versus 45, 50, 55, 60 20 versus 45, 50, 55, 60 25 versus 45, 50, 55, 60 | 10 versus 55, 60 15 versus 45, 50, 55, 60 20 versus 45, 50, 55, 60 25 versus 45, 50, 55, 60 |
| main effect of BASELINE | | $F_{1,436} = 64.583$, $p < 0.001$, partial $\eta^2 = 0.129$ | $F_{1,436} = 45.105$, $p < 0.001$, partial $\eta^2 = 0.098$ | $F_{1,436} = 51.671$, $p < 0.001$, partial $\eta^2 = 0.106$ | $F_{1,413} = 47.188$, $p < 0.001$, partial $\eta^2 = 0.103$ |
| interaction SITE×TIME | | $F_{11,436} = 0.643$, $p = 0.792$, partial $\eta^2 = 0.016$ | $F_{11,413} = 0.603$, $p = 0.826$, partial $\eta^2 = 0.016$ | $F_{11,436} = 1.296$, $p = 0.224$, partial $\eta^2 = 0.032$ | $F_{11,413} = 1.329$, $p = 0.206$, partial $\eta^2 = 0.034$ |

paired t-test

| | log-transformed and normalized pain ratings |
|---|---|
| paired t-test | $t_{19} = 3.903$, $p < 0.01$ |
| mean + 95% CI   control site | 0.7826 (0.6503–0.9150) |
| mean + 95% CI   HFS site | 0.9371 (0.8637–1.011) |
| difference between sites (mean + 95% CI) | 0.1545 (0.0717–0.2373) |

post hoc paired t-tests

| time point (min) | 5 | 10 | 15 | 20 | 25 | 30 | 35 | 40 | 45 | 50 | 55 | 60 |
|---|---|---|---|---|---|---|---|---|---|---|---|---|
| $t_{19}$ | 2.425 | 2.602 | 3.228 | 3.553 | 3.283 | 4.431 | 3.701 | 3.632 | 3.868 | 3.966 | 2.652 | 2.079 |
| $p$ | <0.05 | <0.05 | <0.01 | <0.01 | <0.01 | <0.001 | <0.01 | <0.01 | <0.01 | <0.001 | <0.05 | ns |

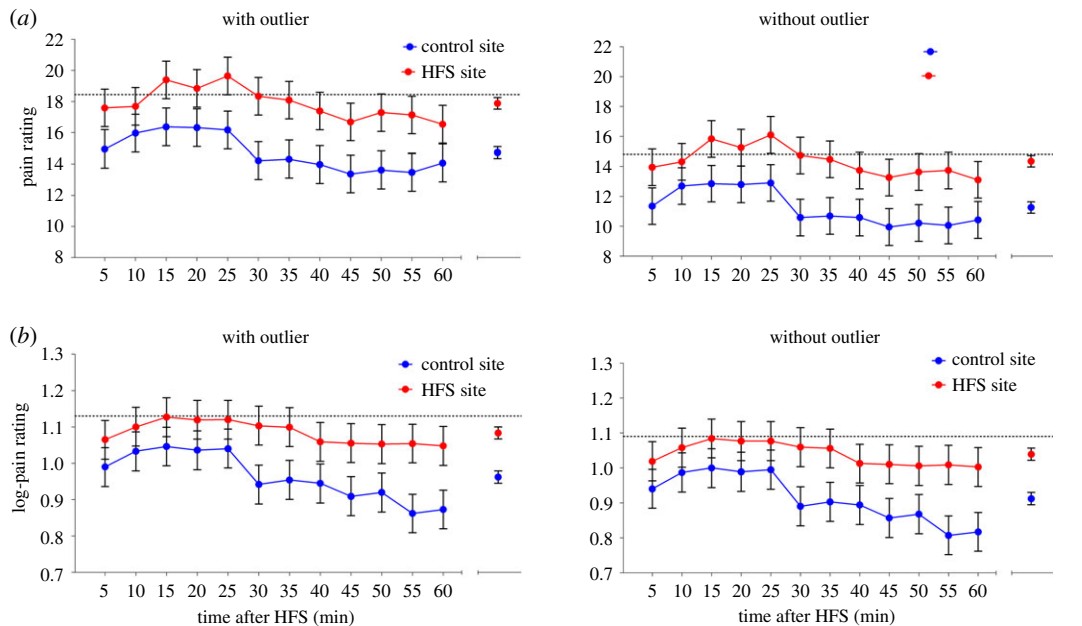

**Figure 4.** Estimated marginal means and 95% confidence interval of the observed pain ratings (*a*) and log-transformed pain ratings (*b*) elicited by the single electrical stimuli after HFS (corrected for baseline) for the two sites (HFS versus control) for the scenarios with and without outlier. At the right side of each figure, the estimated marginal means (and 95% confidence interval) across all time points are shown. The dotted line represents the baseline value.

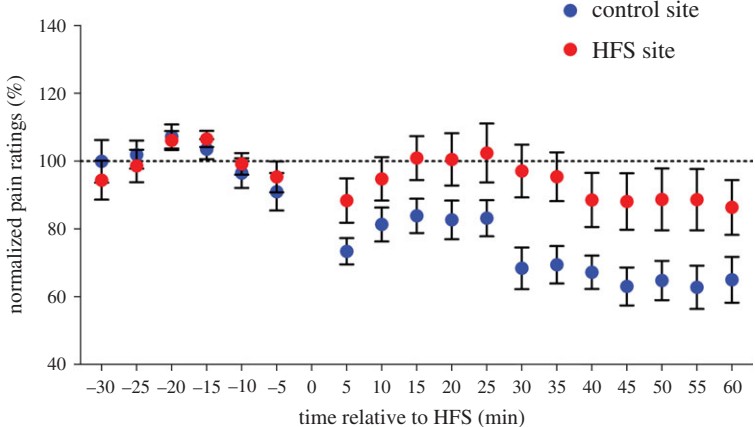

**Figure 5.** Pain ratings to electrical stimuli normalized to baseline. Values represent mean ± s.e.m. Dotted line indicates the mean level of the baseline.

showed a significant difference between the two sites at all but one time points (table 1). Figure 5 depicts the pain ratings normalized to baseline.

## 3.2. Secondary outcome: perceived pain intensity elicited by mechanical pinprick stimuli after high-frequency stimulation

The individual pain ratings elicited by the mechanical pinprick stimuli before and after HFS at the HFS-treated site and control site are shown in figure 6*a*, along with their median and interquartile ranges. Pain ratings were pooled across the two stimulation intensities. Figure 6*b* shows the median and interquartile ranges of the perceived pain intensity without outlier. We conducted only the ANCOVA analyses on the pain ratings elicited by the mechanical pinprick stimuli, given that the assessment of changes in mechanical pinprick sensitivity after HFS was a secondary outcome measure.

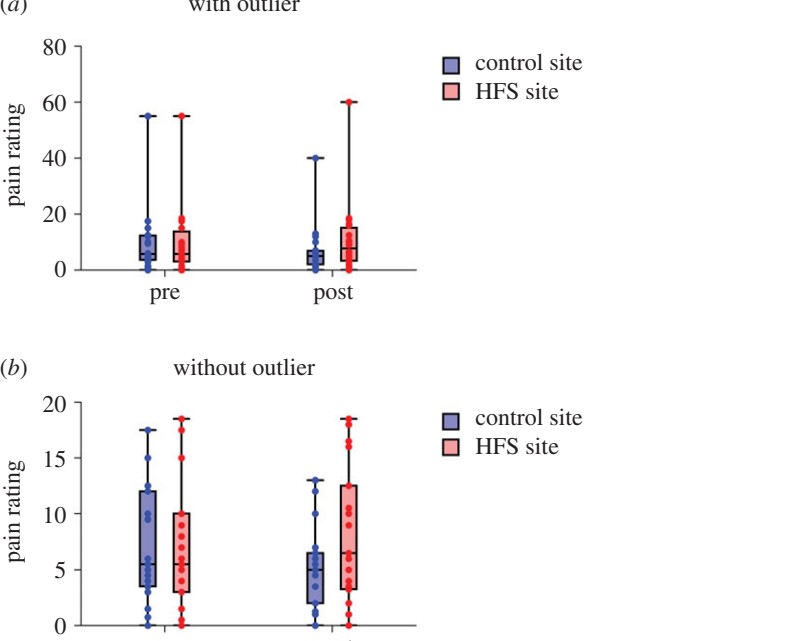

**Figure 6.** Median and interquartile ranges (and min–max value) of the perceived pain intensity elicited by the mechanical pinprick stimuli (pooled across the two stimulation intensities) delivered before and after HFS at the skin surrounding the site at which HFS was applied and at the homologue area of the contra-lateral control site. Dots represent the ratings of each participant.

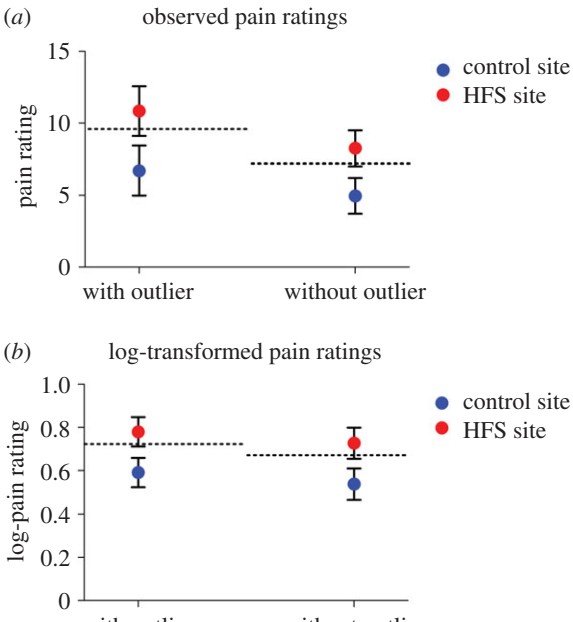

**Figure 7.** Estimated marginal means and 95% confidence interval of the observed pain ratings (*a*) and log-transformed pain ratings (*b*) elicited by mechanical pinprick stimuli (pooled across stimulation intensities) after HFS (corrected for baseline) for the two sites (HFS versus control) for the scenarios with and without outlier. The dotted line represents the baseline values.

### 3.2.1. Repeated measures analysis of covariance

Figure 7 shows the estimated marginal means (and 95% CI) of the perceived pain intensity at both sites after HFS and corrected for pre-existing baseline differences for all four ANCOVA analyses. Table 2 summarizes the main results of the ANCOVA analyses.

**Table 2.** Statistical results for the pain ratings elicited by the mechanical pinprick stimuli.

| ANCOVA | | observed pain ratings with outlier | observed pain ratings without outlier | log-transformed pain ratings with outlier | log-transformed pain ratings without outlier |
|---|---|---|---|---|---|
| main effect of SITE | | $F_{1,18} = 12.494$, $p < 0.01$, partial $\eta^2 = 0.410$ | $F_{1,17} = 15.539$, $p < 0.01$, partial $\eta^2 = 0.478$ | $F_{1,18} = 16.192$, $p < 0.01$, partial $\eta^2 = 0.474$ | $F_{1,17} = 14.561$, $p < 0.01$, partial $\eta^2 = 0.461$ |
| | DIFFERENCE between sites (mean + 95% CI) | 4.125 (1.673–6.577) | 3.289 (1.529–5.050) | 0.189 (0.090–0.287) | 0.189 (0.085–0.294) |
| main effect of BASELINE | | $F_{1,18} = {<}0.001$, $p = 0.991$, partial $\eta^2 < 0.001$ | $F_{1,17} = {<}0.001$, $p = 0.989$, partial $\eta^2 < 0.001$ | $F_{1,18} = 0.014$, $p = 0.908$, partial $\eta^2 < 0.01$ | $F_{1,17} = 0.012$, $p = 0.915$, partial $\eta^2 < 0.01$ |

All analyses showed a significant main effect of 'site', revealing that the perceived pain intensity elicited by the mechanical pinprick stimuli at the HFS-treated site was significantly higher compared to the control site.

# 4. Discussion

The primary aim of this study was to assess if pain elicited by single electrical stimuli was increased after HFS at the conditioned site. We found that pain ratings were higher at the HFS-treated site compared to the control site, independently of the type of statistical analysis we used. However, the difference in pain ratings was mainly due to a decrease of pain at the control site rather than an increase in pain at the HFS-treated site. We thus partially replicated the homotopic effect of HFS reported by Klein *et al*. [3]. As expected, we also found higher pain ratings elicited by mechanical pinprick stimuli after HFS at the skin adjacent to the site at which HFS was delivered compared to the control site (heterotopic effect).

Our finding that the difference in perceived pain intensity elicited by the single electrical stimuli after HFS was mainly due to a decrease of pain at the control site rather than an increase in pain at the HFS-treated site is in contrast with Klein *et al*. [3], and subsequent papers of the same group [8], but in agreement with Matre *et al*. [35]. Indeed, Klein and colleagues consistently found an increase in pain after HFS at the HFS-treated site, whereas Matre *et al*. found a difference in pain ratings after HFS between the two sites that was mainly due to a decrease of pain at the control site rather than an increase in pain at the HFS-treated site. The fact that Klein *et al*. [3] found an increase in pain after HFS at the HFS-treated site cannot be explained by the stimulation intensity of the single electrical stimuli as these were similar. The decrease in pain after HFS at the control site is consistent across all studies and may reflect the phenomenon of habituation. The smaller decrease in pain after HFS at the HFS-treated site, as found by Matre *et al*. [35] and the present study, could be interpreted according to the dual process theory as a mixture of both habituation and sensitization [40].

## 4.1. Mechanisms of high-frequency stimulation-induced changes in pain perception

Hansen *et al*. [7] found that the increase in pain elicited by single electrical stimuli after HFS is characterized predominantly by two sets of descriptors: hot/burning and piercing/stinging, whereby the descriptors hot and burning (and scalding) accounted for more than 40% the variance. The authors suggested that the increase in hot and burning pain is mediated by C-fibres and reflects the perceptual correlate of homosynaptic LTP, an interpretation that would be compatible with the results of animal studies. According to the authors, homosynaptic LTP at Aδ-fibres is unlikely because the activation of Aδ-fibres

by HFS is not sufficient to induce LTP in spinal nociceptive pathways but rather induces long-term depression (LTD; see Hansen et al. [7] for references). Therefore, the authors suggested that the increase in piercing and stinging pain is mediated by Aδ-fibres and reflects the perceptual correlate of heterosynaptic LTP [7].

The increase in mechanical pinprick sensitivity in the heterotopic skin is the most replicable effect of HFS conditioning in humans, although the extent of the increase in the present study was smaller compared to our previous experiments. A possible explanation could be that the increase in pinprick sensitivity is dependent on the stimulation intensity during HFS [3]. Indeed, we usually obtain higher electrical detection thresholds, compared to the thresholds of the present study, and we use a stimulation intensity of 20 times the electrical threshold for delivering HFS [16,17,19,21,27,28]. The increase in pain elicited by mechanical pinprick stimuli delivered to the area adjacent to the site at which HFS was delivered is thought to be mediated by Aδ-fibres and to reflect a perceptual correlate of heterosynaptic LTP [3,8]. Moreover, studies using quantitative sensory testing to assess the changes in perception to natural stimuli after HFS in both the area at which HFS is delivered and the surrounding unconditioned skin predominantly show changes in the perception to mechanical stimuli [6,8] that are highly correlated between the two sites [8], suggesting that heterosynaptic LTP dominates in the area at which HFS is delivered [6,8].

A recent animal study showed that HFS also triggers heterosynaptic LTP at remote C-fibres by activating glial cells that release pro-inflammatory mediators [41]. Moreover, we have shown that in humans the perception elicited by $CO_2$ laser stimuli, which selectively activate cutaneous C-fibres, is enhanced after HFS compared to control site when these stimuli are delivered to the area adjacent to the HFS-treated site [17,27]. These results, together with the notion that heterosynaptic LTP dominates, raise the question if the increase in perceived pain elicited by single electrical stimuli at the HFS-treated site (homotopic area) after HFS is a perceptual correlate of heterosynaptic—rather than homosynaptic—LTP at C-fibres.

# 5. Conclusion

The present study found a significant difference in perceived pain intensity elicited by single electrical stimuli after HFS between the HFS-treated and control site. This difference was mainly due to a decrease of perceived pain at the control site (possibly due to habituation) rather than an increase in pain at the HFS-treated site. We thus partially replicated the homotopic effect of HFS reported by Klein et al. [3]. Whether the lack of decrease in perceived pain at the HFS-treated site is a perceptual correlate of homosynaptic or heterosynaptic changes remains to be investigated.

Ethics. The study was approved by the KU Leuven ethical commission for the Social Sciences (Sociaal-Maatschappelijk Ethische Commissie, SMEC, KU Leuven: G-2020 03 1999). All participants read and signed the informed consent and the exclusion criteria before taking part in the experiment.

Data accessibility. Data are deposited on Dryad at https://doi.org/10.5061/dryad.p8cz8w9nh and the Open Science Framework. Stage 1 pre-registration can be found at https://osf.io/b674f/ (registered on 21 September 2020). This pre-registration was undertaken prior to data collection and analysis.

Authors' contributions. E.N.v.d.B. and D.M.T. conceived the experiment; E.N.v.d.B., T.V., J.B.-M and D.M.T. discussed the set-up, analysed the data and drafted the manuscript; T.V. acquired the data; all authors provided critical inputs on the content of the article, revised it and gave their final approval.

Competing interests. We declare we have no competing interests.

Funding. E.N.v.d.B. is supported by the Fonds de Recherche Clinique (Clinique Universitaire St Luc) and by the Queen Elisabeth Medical Foundation for Neuroscience (Young Researchers grant); D.M.T. is supported by a BOFZAP Starting Grant (KU Leuven), by the 'Asthenes' Methusalem grant by the Flemish Government, Belgium and by a Research Grant (Junior Project) by the FWO.

Acknowledgements. We acknowledge the help of Mathijs Franssen in the preparation of the set-up. We also thank Thomas Klein, one of the original authors, for answering all our questions about the experimental design.

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
