## [Reviewer comments · Royal Society Open Science]

Review History

RSOS-200830.R0 (Original submission)

Review form: Reviewer 1

Do you have any ethical concerns with this paper?

No

Have you any concerns about statistical analyses in this paper?

No

Recommendation?

Accept in principle

Comments to the Author(s)

I think this replication is as precise as it can be. Sample size calculation is convincing.

Review form: Reviewer 2

Do you have any ethical concerns with this paper?

No

Have you any concerns about statistical analyses in this paper?

No

Recommendation?

Accept in principle

Comments to the Author(s)

Nicely described protocol, I wish you good luck with the study.

Review form: Reviewer 3 (Carsten Dahl Mørch)

Do you have any ethical concerns with this paper?

No

Have you any concerns about statistical analyses in this paper?

Yes

Recommendation?

Accept with minor revision

Comments to the Author(s)

In this stage 1 version of the paper entitled "Perceptual correlates of homosynaptic long term potentiation in human nociceptive pathways: a replication study" van den Broeke et al proposes a replication study of "Perceptual Correlates of Nociceptive Long-Term Potentiation and Long-Term Depression in Humans" by Klien et al. It is appropriate to attempt a close replication of this study. Several research groups have used the 'Klein' model of Pain-LTP, but usually with small methodological difference and some differences in the results. It seems consistent to observe increased response to mechanical pin prick stimulation across studies, but the response electrical stimulation seems rarely to increase as originally reported by Klein et al.

The methods of this replication study are clearly described to follow to original article very closely. I do have a few minor concerns:

- 1) The number of mechanical pinprick stimuli is substantially lower the replication study. Now, the main aim of the reproducibility study is to assess the electrical response, but the authors should justify how it can be excluded that the mechanical stimulation is not part of the maintenance of LTP.
- 2) The statistical analysis seems to differ from the original study. The original study normalizes to the baseline by division: "Except for touch-evoked pain, ratings were normalized to baseline by dividing postconditioning pain ratings by the mean value of the 40 min baseline period", while the replication study normalizes by adding the mean of the baseline as a covariate. While I would also prefer the covariant solution, this will act as a subtraction and thus differ from the original study.
- 3) The sample size estimation should be explained in further detail, as the methods is not standard to most readers. The original study had 8 participants, why was n0 set to 7?

Decision letter (RSOS-200830.R0)

Dear Professor Torta

On behalf of the Editors, I am pleased to inform you that your Manuscript RSOS-200830 entitled "Perceptual correlates of homosynaptic long term potentiation in human nociceptive pathways: a replication study" deemed suitable for in-principle acceptance in Royal Society Open Science subject to minor revision in accordance with the referee and editor suggestions. Please find their comments at the end of this email.

The reviewers and handling editors have recommended publication, but also suggest some minor revisions to your manuscript. Therefore, I invite you to respond to the comments and revise your manuscript.

Please you submit the revised version of your manuscript within 7 days (i.e. by the 25-Jun-2020). If you do not think you will be able to meet this date please let me know immediately.

When submitting your revised manuscript, you will be able to respond to the comments made by the referees and upload a file "Response to Referees" in the "File Upload" step. You can use this to document any changes you make to the original manuscript. In order to expedite the processing of the revised manuscript, please be as specific as possible in your response to the referees.

Full author guidelines can be found here <https://royalsocietypublishing.org/rsos/replication-studies#AuthorsGuidance>.

Kind regards,
Andrew Dunn
Royal Society Open Science
openscience@royalsociety.org

on behalf of Professor Chris Chambers (Registered Reports Editor, Royal Society Open Science)
openscience@royalsociety.org

Associate Editor Comments to Author (Professor Chris Chambers):

Associate Editor: 1

Comments to the Author:

Three specialist reviewers have now appraised the manuscript. All are positive, in fact, two of the reviewers recommend acceptance with no revisions. Reviewer 3, however, notes several areas

requiring attention before IPA can be awarded, including deviations from the original protocol and clarification of the sampling plan. Please attend carefully to these points in a response and revised manuscript.

Reviewers' comments to Author:

Reviewer: 1

Comments to the Author(s)

I think this replication is as precise as it can be. Sample size calculation is convincing.

Reviewer: 2

Comments to the Author(s)

Nicely described protocol, I wish you good luck with the study.

Reviewer: 3

Comments to the Author(s)

IN this stage 1 version of the paper entitled "Perceptual correlates of homosynaptic long term potentiation in human nociceptive pathways: a replication study" van den Broeke et al proposes a replication study of "Perceptual Correlates of Nociceptive Long-Term Potentiation and Long-Term Depression in Humans" by Klien et al. It is appropriate to attempt a close replication of this study. Several research groups have used the 'Klein' model of Pain-LTP, but usually with small methodological difference and some differences in the results. It seems consistent to observe increased response to mechanical pin prick stimulation across studies, but the response electrical stimulation seems rarely to increase as originally reported by Klein et al.

The methods of this replication study are clearly described to follow to original article very closely. I do have a few minor concerns:

- 1) The number of mechanical pinprick stimuli is substantially lower the replication study. Now, the main aim of the reproducibility study is to assess the electrical response, but the authors should justify how it can be excluded that the mechanical stimulation is not part of the maintenance of LTP.
- 2) The statistical analysis seems to differ from the original study. The original study normalizes to the baseline by division: "Except for touch-evoked pain, ratings were normalized to baseline by dividing postconditioning pain ratings by the mean value of the 40 min baseline period", while the replication study normalizes by adding the mean of the baseline as a covariate. While I would also prefer the covariant solution, this will act as a subtraction and thus differ from the original study.
- 3) The sample size estimation should be explained in further detail, as the methods is not standard to most readers. The original study had 8 participants, why was n0 set to 7?

Author's Response to Decision Letter for (RSOS-200830.R0)

See Appendix A.

Decision letter (RSOS-200830.R1)

Dear Professor Torta

On behalf of the Editor, I am pleased to inform you that your Manuscript RSOS-200830.R1 entitled "Perceptual correlates of homosynaptic long term potentiation in human nociceptive pathways: a replication study" has been accepted in principle for publication in Royal Society Open Science.

You may now progress to Stage 2 and complete the study as approved.

Please note that you must now register your approved protocol on the Open Science Framework (<https://osf.io/rr>), using the 'Submit your approved Registered Report' option and then the 'Registered Report Protocol Preregistration' option. Please use the Registered Report option even though your article is being accepted as a Stage 1 Replication. Further into the registration process, in the Journal Title field enter 'Royal Society Open Science (Replication article type, Preregistered track)'. Please note that a time-stamped, independent registration of the protocol is mandatory under journal policy, and manuscripts that do not conform to this requirement cannot be considered at Stage 2. The protocol should be registered unchanged from its current approved state. Please include a URL to the protocol in your Stage 2 manuscript (e.g. 'This article received in-principle acceptance (IPA) at Royal Society Open Science on 24 June 202. Following IPA, the accepted Stage 1 version of the manuscript was preregistered on the OSF (URL). This preregistration was performed prior to data collection and analysis.')

We would be grateful if you could now update the journal office as to the anticipated completion date of your study.

Following completion of your study, we invite you to resubmit your paper for peer review as a Stage 2 Replication. Please note that your manuscript can still be rejected for publication at Stage 2 if the Editors consider any of the following conditions to be met:

- The Introduction and methods deviated from the approved Stage 1 submission (required).
- The authors' conclusions were not considered justified given the data.

We encourage you to read the complete guidelines for authors concerning Stage 2 submissions at: <https://royalsocietypublishing.org/rsos/replication-studies#AuthorsGuidance>. Please especially note the requirements for data sharing and that withdrawing your manuscript will result in publication of a Withdrawn Registration.

Once again, thank you for submitting your manuscript to Royal Society Open Science and I look forward to receiving your Stage 2 submission. If you have any questions at all, please do not hesitate to get in touch. We look forward to hearing from you shortly with the anticipated submission date for your stage two manuscript.

Kind regards,
Andrew Dunn
Royal Society Open Science
openscience@royalsociety.org

on behalf of Professor Chris Chambers (Registered Reports Editor, Royal Society Open Science)
openscience@royalsociety.org

Author's Response to Decision Letter for (RSOS-200830.R1)

See Appendix B.

RSOS-200830.R2 (Revision)

Review form: Reviewer 2

Is the manuscript scientifically sound in its present form?

Yes

Is the language acceptable?

Yes

Do you have any ethical concerns with this paper?

No

Have you any concerns about statistical analyses in this paper?

No

Recommendation?

Accept with minor revision

Comments to the Author(s)

Intro and methods

- New from the Stage 1 submission is that the authors performed a multiverse analysis, i.e. different analyses and displaying how stable and robust the findings are. This approach is new to me, but I find it appealing. Ideally, this approach should have been described a priori at Stage 1. Deciding on new statistical approaches after data collection may lead to p hacking. I don't insinuate that this happened here, however.
- Statistics section 3.7: Please explain post hoc tests comparing time points, were all comparisons made? From Table 1, it seems like time 15, 20 and 25 were compared with the later time points. What was the rationale for selecting these three in particular?

Results

- I like the honest presentation of pain ratings in box plots (Figure 2), with and without outliers.
- P8, line 37, section 4.1.1: Please explain "corrected for pre-existing baseline differences". Was baseline pain ratings subtracted from post-HFS pain ratings? If so, were the mean baseline pain ratings across sites calculated? Is this the same baseline as in figure 4, dotted line? Please explain this in the start of section 3.7 or in a new Data analysis section.

Review form: Reviewer 3 (Carsten Dahl Mørch)

Is the manuscript scientifically sound in its present form?

Yes

Is the language acceptable?

Yes

Do you have any ethical concerns with this paper?

No

Have you any concerns about statistical analyses in this paper?

No

Recommendation?

Accept with minor revision

Comments to the Author(s)

4.1 primary outcome.

The analysis of the pain rating response to single electrical stimulation showed an outlier reporting very high pain intensities. The removal of the outlier and reanalysis is appreciative but not really necessary. A repeated measures ANCOVA is used for analysis, thus adding a term for the pain intensity for each subject to analysis by the 'repeated' setup. The average preconditioning ratings were further used as a normalization factor. These two normalization approaches assume linearity of the effects, and will differ when log-transform is applied. I would recommend that the authors only deviated from the planned analysis if the residuals of the planned analysis give any indications for log transforming and outlier removal. From table 1, it seems that the 4 analysis gives quite similar results.

Table 1 is quite busy e.g. with multiple t-tests. Please consider two tables.

From figure 5 it seems clear that the pain ratings are not increased by HFS but rather that the control site is decreased. The statement from the abstract is of course true: "Furthermore, we found a higher perceived pain intensity for single electrical stimuli delivered to the HFS-treated site compared to the control site" but the indication that this is a homotopic effect is rather speculative. I suggest to remove the "(homotopic effect)" statement. The discussion on this subject is clear and appropriate.

Typo in page 8 line 59: "but we not cannot be"

Page 9 line 32 "Indeed, we usually obtain" please provide references for these studies, and preferably studies from other groups as well.

Decision letter (RSOS-200830.R2)

Dear Professor Torta

On behalf of the Editor, I am pleased to inform you that your Stage 2 Replication submission RSOS-200830.R2 entitled "Perceptual correlates of homosynaptic long term potentiation in human nociceptive pathways: a replication study" has been accepted for publication in Royal Society Open Science subject to minor revision in accordance with the referee suggestions. Please find the referees' comments at the end of this email.

The reviewers and Subject Editor have recommended publication, but also suggest some minor revisions to your manuscript. Therefore, I invite you to respond to the comments and revise your manuscript.

Please also ensure that all the below editorial sections are included where appropriate (a non-exhaustive example is included in an attachment):

- Ethics statement

- Data accessibility

If you wish to submit your supporting data or code to Dryad (<http://datadryad.org/>), or modify your current submission to dryad, please use the following link:
<http://datadryad.org/submit?journalID=RSOS&manu=RSOS-200830.R2>

- Competing interests

- Authors' contributions

- Acknowledgements

- Funding statement

Because the schedule for publication is very tight, it is a condition of publication that you submit the revised version of your manuscript within 7 days (i.e. by the 11-Dec-2020). If you do not think you will be able to meet this date please let me know immediately.

- 1) A text file of the manuscript (tex, txt, rtf, docx or doc), references, tables (including captions) and figure captions. Do not upload a PDF as your "Main Document".
- 2) A separate electronic file of each figure (EPS or print-quality PDF preferred (either format should be produced directly from original creation package), or original software format)
- 3) Included a 100 word media summary of your paper when requested at submission. Please ensure you have entered correct contact details (email, institution and telephone) in your user account
- 4) Included the raw data to support the claims made in your paper. You can either include your data as electronic supplementary material or upload to a repository and include the relevant DOI within your manuscript
- 5) Included your supplementary files in a format you are happy with (no line numbers, Vancouver referencing, track changes removed etc) as these files will NOT be edited in production

Kind regards,
Professor Chris Chambers
Royal Society Open Science
openscience@royalsociety.org

on behalf of Professor Chris Chambers (Registered Reports Editor, Royal Society Open Science)
openscience@royalsociety.org

Associate Editor Comments to Author (Professor Chris Chambers):

Associate Editor: 1

Comments to the Author:

Two of the three reviewers who assessed the Stage 1 submission were available to assess the Stage 2 submission (Reviewer 2 and Reviewer 3). As you will see, both are broadly positive about

the completed manuscript and offer some helpful comments for clarifying aspects of the analysis and general presentation. Concerning the comment of Reviewer 2 that the multiverse analysis should have been prespecified at Stage 1, it is of course the case that this plan *was* prespecified in the revised (and preregistered) Stage 1 manuscript (in response to the Stage 1 review provided by Reviewer 3), and as linked within the the Stage 2 manuscript; however it is understandable that Reviewer 2 would have been unaware of this addition to the protocol at the time because, given the minor nature of the Stage 1 revision, IPA was offered without returning the Stage 1 manuscript to the reviewers. You therefore need not respond to this specific point of the reviewer and the comment about risk of p-hacking.

Please attend carefully to all remaining comments. Concerning the suggested re-analysis suggested by Reviewer 3, please respond appropriately while remaining true to the preregistered protocol.

Reviewers' comments to Author:

Reviewer: 2

Comments to the Author(s)

Intro and methods

- New from the Stage 1 submission is that the authors performed a multiverse analysis, i.e. different analyses and displaying how stable and robust the findings are. This approach is new to me, but I find it appealing. Ideally, this approach should have been described a priori at Stage 1. Deciding on new statistical approaches after data collection may lead to p hacking. I don't insinuate that this happened here, however.
- Statistics section 3.7: Please explain post hoc tests comparing time points, were all comparisons made? From Table 1, it seems like time 15, 20 and 25 were compared with the later time points. What was the rationale for selecting these three in particular?

Results

- I like the honest presentation of pain ratings in box plots (Figure 2), with and without outliers.
- P8, line 37, section 4.1.1: Please explain "corrected for pre-existing baseline differences". Was baseline pain ratings subtracted from post-HFS pain ratings? If so, were the mean baseline pain ratings across sites calculated? Is this the same baseline as in figure 4, dotted line? Please explain this in the start of section 3.7 or in a new Data analysis section.

Reviewer: 3

Comments to the Author(s)

4.1 primary outcome.

The analysis of the pain rating response to single electrical stimulation showed an outlier reporting very high pain intensities. The removal of the outlier and reanalysis is appreciative but not really necessary. A repeated measures ANCOVA is used for analysis, thus adding a term for the pain intensity for each subject to analysis by the 'repeated' setup. The average preconditioning ratings were further used as a normalization factor. These two normalization approaches assume linearity of the effects, and will differ when log-transform is applied. I would recommend that the authors only deviated from the planned analysis if the residuals of the planned analysis give any indications for log transforming and outlier removal. From table 1, it seems that the 4 analysis gives quite similar results.

Table 1 is quite busy e.g. with multiple t-tests. Please consider two tables.

From figure 5 it seems clear that the pain ratings are not increased by HFS but rather that the control site is decreased. The statement from the abstract is of course true: "Furthermore, we found a higher perceived pain intensity for single electrical stimuli delivered to the HFS-treated

site compared to the control site" but the indication that this is a homotopic effect is rather speculative. I suggest to remove the "(homotopic effect)" statement. The discussion on this subject is clear and appropriate.

Typo in page 8 line 59: "but we not cannot be"

Page 9 line 32 "Indeed, we usually obtain" please provide references for these studies, and preferably studies from other groups as well.

Author's Response to Decision Letter for (RSOS-200830.R2)

See Appendix C.

Decision letter (RSOS-200830.R3)

This year has been very difficult for everyone, and we want to take the opportunity to thank you for your continued support in 2020.

The Royal Society Open Science editorial office will be closed from the evening of Friday 18 December 2020 until Monday 4 January 2021. We will not be responding during this time. If you have received a deadline within this time period, please contact us as soon as possible to allow us to extend the deadline. If you receive any automated messages during this time asking you to meet a deadline, we offer apologies and invite you to respond after the festive period or during normal working hours.

With our best for a peaceful festive period and New Year, and we look forward to working with you in 2021.

Dear Professor Torta:

It is a pleasure to accept your manuscript entitled "Perceptual correlates of homosynaptic long term potentiation in human nociceptive pathways: a replication study" in its current form for publication in Royal Society Open Science.

You can expect to receive a proof of your article in the near future, though please bear in mind that this may be in early 2021 owing to the festive break. Please contact the editorial office (openscience@royalsociety.org) and the production office (openscience_proofs@royalsociety.org) to let us know if you are likely to be away from e-mail contact -- if you are going to be away, please nominate a co-author (if available) to manage the proofing process, and ensure they are copied into your email to the journal.

on behalf of Professor Chris Chambers (Subject Editor)
openscience@royalsociety.org

Appendix A

Associate Editor Comments to Author (Professor Chris Chambers):
Associate Editor: 1

Comments to the Author:

“Three specialist reviewers have now appraised the manuscript. All are positive, in fact, two of the reviewers recommend acceptance with no revisions. Reviewer 3, however, notes several areas requiring attention before IPA can be awarded, including deviations from the original protocol and clarification of the sampling plan. Please attend carefully to these points in a response and revised manuscript.”

We thank all positive comments and appreciation for our intention. We have addressed reviewer 3's comments both here and in the manuscript.

Reviewer:

3

Comments to the Author(s)

IN this stage 1 verison of the paper entitled “Perceptual correlates of homosynaptic long term potentiation in human nociceptive pathways: a replication study” van den Broeke et al proposes a replication study of “Perceptual Correlates of Nociceptive Long-Term Potentiation and Long-Term Depression in Humans” by Klien et al. It is appropriate to attempt a close replication of this study. Several research groups have used the ‘Klein’ model of Pain-LTP, but usually with small methodological difference and some differences in the results. It seems consistent to observe increased response to mechanical pin prick stimulation across studies, but the response electrical stimulation seems rarely to increase as originally reported by Klein et al. The methods of this replication study are clearly described to follow to original article very closely. I do have a few minor concerns:

The number of mechanical pinprick stimuli is substantially lower the replication study. Now, the main aim of the reproducibility study is to assess the electrical response, but the authors should justify how it can be excluded that the mechanical stimulation is not part of the maintenance of LTP.

We thank the reviewer for raising this point. We do not believe that this is a main concern, in fact in the Hansen et al., paper (2007), Hansen, Klein, Magerl and Treede show that homotopic LTP can be observed without the measurement of pinprick sensitivity. We have added a sentence clarifying this point at page 3. Here is how the text currently reads:

*“[...] Alternating between mechanical stimuli and electrical ones may bias the judgement of intensity of electrical stimuli towards higher ratings, **although available evidence seems to advocate against this possibility [7]**”*

2) *The statistical analysis seems to differ from the original study. The original study normalizes to the baseline by division: “Except for touch-evoked pain, ratings were normalized to baseline by dividing postconditioning pain ratings by the mean value of the 40 min baseline period”, while the replication study normalizes by adding the mean of the baseline as a covariate. While I would also prefer the covariant solution, this will act as a subtraction and thus differ from the original study.*

A possible solution to adopt the co-variate solution, and yet to take into account the statistical differences is to perform a multiverse analysis. Multiverse analysis offers the possibility of testing multiple possible statistical choices and provides an idea of how much such choices impact on the observed results.

This part has been added to the manuscript at page 5, and reads:

“Since our statistical approach differed from that of the original paper, we performed a multiverse analysis. This approach involves performing different analysis across reasonable choices for data processing, and displays how stable and robust the findings are [40]”.

3) *The sample size estimation should be explained in further detail, as the methods is not standard to most readers. The original study had 8 participants, why was n0 set to 7?*

We set the n0 to 7 as one of the participants was discarded from the analysis of the homotopic. It is clear from the degrees of freedom reported in the ANOVA table. We have reported the original table from Klein et al., 2005 here.

We have better specified this point in the sample size paragraph.

In this case, $p_0=0.0023$ and $n_0=7$ (for the homotopic analysis, see [3] table 1), which resulted in a relative sample size $c=2.5$. Therefore, we recruited 20 participants.

Table 1. ANOVA for pain ratings after conditioning stimulation

	HFS			LFS		
	df	F	p	df	F	p
Homotopic effects (electrical; n = 7)						
Conditioned versus control	1,6	25.97	p < 0.01	1,6	25.86	p < 0.01
Conditioning stimulus intensity	1,6	0.19	p = 0.68	1,6	2.15	p = 0.19
Interaction	1,6	0.28	p = 0.62	1,6	1.67	p = 0.24
Heterotopic effects (pin prick; n = 8)						
Conditioned versus control	1,7	54.73	p < 0.001	1,7	8.34	p < 0.05
Conditioning stimulus intensity	1,7	1.18	p = 0.31	1,7	11.85	p < 0.05
Interaction	1,7	1.8	p = 0.22	1,7	7.07	p < 0.05
Heterotopic effects (light touch; n = 8)						
Conditioned versus control	1,7	8.65	p < 0.05			
Conditioning stimulus intensity	1,7	2.49	p = 0.16			
Interaction	1,7	2.49	p = 0.16			

Allodynia after LFS was observed only in one subject at 20 × T; thus the statistical analysis appeared inadequate.

Dear Editors,

Herewith, we are submitting our Stage 2 manuscript entitled "*Perceptual correlates of homotopic long term potentiation in nociceptive pathways: a replication study*" by van den Broeke EN, Vanmaele T, Mouraux A, Stouffs A, Biurrun-Manresa J, Torta DM.

In the Stage 2 findings, we have found a significant difference of the perceived pain intensity elicited by single electrical stimuli after HFS between the HFS-treated site and control site. However, in contrast to the results reported in the original paper of Klein et al. (2004), but in agreement with the results of Matre et al. (2013), this difference was mainly due to a decrease of pain at the control site rather than an increase of pain at the HFS-treated site. We thus *partially* replicated the homotopic effect of HFS reported by Klein et al. (2004). Our multiverse analysis showed that this result was independent of the statistical procedure that was used.

We have to mention two minor deviations from our original protocol but which do not affect the results of our primary outcome. The first involves the use of a newer version of the electrical stimulator than the one that is mentioned in the protocol. This because the initial stimulator was already booked intensively by another group in our laboratory. The second deviation involves our decision to not ask for pain ratings *during* HFS, after each train. During the piloting we noticed that asking a rating after each HFS train, could interfere with the HFS stimulation as some subjects were rather late in reporting their rating, which had as consequence that when they reported their rating the next train had already arrived. We thought therefore, in order to not introduce any confounding effect of providing a rating at variable times *during* the HFS stimulation that it was better to remove this element from the protocol. Both Xia et al. (2016) and Matre et al. (2013) did ask for ratings during HFS, however, the homotopic effect was either not observed (Xia et al.) or the result was the same as ours (Matre et al.). Therefore, providing a rating *during* HFS does not increase the probability of observing the homotopic effect of HFS. More importantly, homosynaptic LTP induced by HFS is demonstrated in animal studies in which, obviously, no pain ratings during HFS were obtained.

Finally, we would like to mention that we have spotted two errors in the manuscript version accepted at stage 1, which were overlooked by the reviewers as well. In the text of the paragraph describing the statistical analysis we mention as the first time-point of the factor TIME "*directly after HFS*", but this is incorrect, as our first post measurement started at 5 min after applying HFS as shown in Figure 1. Furthermore, in the paragraph "Study design" we mention that HFS was applied to the volar forearm of the dominant arm, but we delivered HFS to the left or right volar forearm as mentioned in Figure 1. We have now corrected these minor errors in the text of the manuscript.

Diana Torta & Emanuel van den Broeke

Emanuel van den Broeke
Institute of Neuroscience, UC Louvain
Avenue Mounier 53, B-1200 Brussels
emanuelvandenbroeke@uclouvain.be

Diana M. Torta
Research Group on Health Psychology, KU Leuven
Tiensestraat 102, B-3000 Leuven
diana.torta@kuleuven.be

Appendix C

Responses to the Reviewers

Reviewer #2

Intro and methods

- Statistics section 3.7: Please explain post hoc tests comparing time points, were all comparisons made? From Table 1, it seems like time 15, 20 and 25 were compared with the later time points. What was the rationale for selecting these three in particular?

Reply from the authors

In fact, for the main effect of time all comparisons were made. We have now added this to the text of section 3.7.

Results

- P8, line 37, section 4.1.1: Please explain “corrected for pre-existing baseline differences”. Was baseline pain ratings subtracted from post-HFS pain ratings? If so, were the mean baseline pain ratings across sites calculated? Is this the same baseline as in figure 4, dotted line? Please explain this in the start of section 3.7 or in a new Data analysis section.

Reply from the authors

The sentence “corrected for pre-existing baseline differences” refers to the covariate, which was the mean baseline (-30-0 min) pain rating, separately for each site, for each subject. We have now added this to paragraph 3.7.

Reviewer #3

4.1 primary outcome.

The analysis of the pain rating response to single electrical stimulation showed an outlier reporting very high pain intensities. The removal of the outlier and reanalysis is appreciative but not really necessary. A repeated measures ANCOVA is used for analysis, thus adding a term for the pain intensity for each subject to analysis by the ‘repeated’ setup. The average preconditioning ratings were further used as a normalization factor. These two normalization approaches assume linearity of the effects, and will differ when log-transform is applied. I would recommend that the authors only deviated from the planned analysis if the residuals of the planned analysis give any indications for log transforming and outlier removal. From table 1, it seems that the 4 analysis gives quite similar results.

Reply from the authors

We would like to clarify that the multiverse analysis was chosen a priori to show the robustness of the effects. Therefore we show the ANCOVA analysis with and without outlier and with and without log-transform. Importantly, the latter was included because the authors of the original paper (Klein et al. 2004) log-transformed their data, and in order to show the robustness of the findings we showed the ANCOVA with and without log-transform.

Table 1 is quite busy e.g. with multiple t-tests. Please consider two tables.

Reply from the authors

We prefer to keep all the results regarding the homotopic effects in one table. Besides we have already two tables.

From figure 5 it seems clear that the pain ratings are not increased by HFS but rather that the control site is decreased. The statement from the abstract is of course true: “Furthermore, we found a higher perceived pain intensity for single electrical stimuli delivered to the HFS-treated site compared to the control site” but the indication that this is a homotopic effect is rather speculative. I suggest to remove the “(homotopic effect)” statement. The discussion on this subject is clear and appropriate.

Reply from the authors

We thank the reviewer for their positive comments. We do not see why naming the change in pain ratings observed at the site at which HFS was applied as homotopic would be speculative. We prefer to keep the wording “homotopic effect” because it refers to the change in pain ratings observed at the site at which HFS was applied (homotopic) and to contrast to the heterotopic effects (change in pinprick sensitivity of the skin surrounding the site at which HFS was applied).

Typo in page 8 line 59: “but we not cannot be”

Reply from the authors

We have corrected the typo.

Page 9 line 32 “Indeed, we usually obtain” please provide references for these studies, and preferably studies from other groups as well.

Reply from the authors

We now have added references to our studies.